# Single-Walled Carbon Nanohorn-Based Fluorescence Energy Resonance Transfer Aptasensor Platform for the Detection of Aflatoxin B1

**DOI:** 10.3390/foods12152880

**Published:** 2023-07-28

**Authors:** Yiting Fan, Huanhuan Yang, Jiaxin Li, Khalid Amin, Bo Lyu, Wendan Jing, Sainan Wang, Hongling Fu, Hansong Yu, Zhijun Guo

**Affiliations:** 1College of Food Science and Engineering, Jilin Agricultural University, Changchun 130118, China; 2Division of Soybean Processing, Soybean Research & Development Center, Chinese Agricultural Research System, Changchun 130118, China; 3College of Food Science, Heilongjiang Bayi Agricultural University, Daqing 163000, China; 4College of Life Science, Chang Chun Normal University, Changchun 130032, China; 5Nutrition and Bromatology Group, Department of Analytical and Food Chemistry, Faculty of Sciences, Universidade de Vigo, 32004 Ourense, Spain; 6College of Agriculture, Yanbian University, Yanji 133002, China

**Keywords:** aflatoxin B1, aptasensor, forster resonance energy transfer, SWCNHs, rapid detection

## Abstract

Aflatoxin B1 (AFB1) is one of the most contaminated fungal toxins worldwide and is prone to cause serious economic losses, food insecurity, and health hazards to humans. The rapid, on-site, and economical method for AFB1 detection is need of the day. In this study, an AFB1 aptamer (AFB1-Apt) sensing platform was established for the detection of AFB1. Fluorescent moiety (FAM)-modified aptamers were used for fluorescence response and quenching, based on the adsorption quenching function of single-walled carbon nanohorns (SWCNHs). Basically, in our constructed sensing platform, the AFB1 specifically binds to AFB1-Apt, making a stable complex. This complex with fluorophore resists to be adsorbed by SWCNHs, thus prevent SWCNHs from quenching of fluorscence, resulting in a fluorescence response. This designed sensing strategy was highly selective with a good linear response in the range of 10–100 ng/mL and a low detection limit of 4.1 ng/mL. The practicality of this sensing strategy was verified by using successful spiking experiments on real samples of soybean oil and comparison with the enzyme-linked immunosorbent assay (ELISA) method.

## 1. Introduction

Aflatoxin B1 (AFB1) is a common fungal toxin produced by Aspergillus flavus and Aspergillus parasiticus as a toxic secondary metabolite [1] AFB1 contamination is widespread, and is widely present in the growth, transport, storage, and processing of agricultural products. It is one of the major fungal toxins contributing to global food insecurity in a continuesly growing population [2,3]. Toxicological studies have shown that AFB1 is strongly toxic and can cause serious damage to human tissues and organs. It also increases carcinogenicity, teratogenicity, mutagenicity, and immune suppression. That is why it is classified as a Class I carcinogen by the International Agency for Research on Cancer (IARC) [4,5]. Because of its potential threat to economy and human health, the Food and Drug Administration (FDA) and the European Union (EU) have set maximum levels of AFB1 at 20 ppb and 2 ppb in crops and food, respectively [6]. Following the above consequences, quantitative analysis of AFB1 contamination in food raw materials is essential.

In recent years, conventional methods for AFB1 detection are focused on instrumental and immunoassay methods [7]. Among them, instrumental analytical methods include thin layer chromatography (TLC) [8], liquid chromatography-tandem mass spectrometry (LC-MS) [9], and high-performance liquid chromatography (HPLC) [10]. However, these methods and need complicated pre-processing, expensive instrumentation, specialized operators, and long time, which limit their application in rapid detection at the field level [11,12]. In addition, antibody-based immunoassays, such as the enzyme-linked immunosorbent assay (ELISA) [13], immunoaffinity column (IAC) [14], and immunochromatographic strip (ICS) [15], enable the rapid detection of AFB1 with some sensitivity. However, the instability of the antibody and the complicated washing and incubation process are prone to produce false-positive or false-negative results, which limits the practical ability of the immunoassays [16,17]. Therefore, there is an urgent need to develop a simple, efficient, economical, and relatively fast analysis method that can be used as an alternative.

Aptamers are single-stranded non-coding DNA (ssDNA) or RNA (ssRNA) sequences screened in vitro using systematic evolution of exponentially enriched ligands (SELEX) [18,19]. They can recognize and bind to specific target molecules, like proteins, with high specificity and selectivity resulting in a solid tertiary structure [20]. Compared with traditional recognition antibodies, aptamers have the advantages of high stability, easy modification, a wide range of targets, no immunogenicity, and good specificity. Thus aptamers are often considered as the best alternative to antibodies and regarded as best choice among recognition and binding elements [21,22,23,24]. Therefore, aptamer-based biosensors can overcome the above limitations of chromatography and ELISA and provide a more convenient detection strategy for AFB1 detection.

In recent years, nano-materials have widely been combined with target-sensitive and cost-effective analytical tools due to their unique physical properties and chemical stability [25,26]. Among them, carbon nanomaterials have gained much attention in biosensing applications due to their excellent electrical conductivity, large specific surface area, and biocompatibility [27,28,29,30]. Carbon nanohorns (CNHs) are a new type of carbon nanomaterials after carbon nanotubes (CNTs), which are conical structures starting from a pentacyclic structure and expanding outward from a hexagonal graphite structure [31]). In addition to the excellent properties of the original carbon nano-materials, single-walled carbon nanohorns (SWCNHs) have a larger specific surface area and better dispersion, which increases the applicability of SWCNHs [32]. Secondly, SWCNHs can adsorb nucleic acid base loops on fluorophore-labeled ssDNA through hydrophobic interactions, hydrogen bonding, and π-π stacking interactions. Comparatively double-stranded DNA (dsDNA) cannot be adsorbed on the surface of SWCNHs due to the hydrogen bonding between phosphate backbone of nucleic acid bases [33]. It has been successfully demonstrated that, these features allows fluorescent groups and SWCNHs to undergo fluorescence resonance energy transfer (FRET) effects leading to fluorescence bursts [34]. The construction of this sensing strategy promotes the development of aptasensors and provides a simpler, more efficient, and low-cost detection platform for a wide range of researchers.

In this study, we reported a simple, rapid, and low-cost fluorescent aptamer sensing and analyzing method based on SWCNHs for the specific detection of AFB1. The assay mainly consists of fluorescein amidites-modified (FAM-modified) AFB1 aptamer (AFB1-Apt) as a fluorescent probe and SWCNHs as a bursting agent. The unique properties of SWCNHs adsorbing ssDNA nucleic acid aptamers were used to design the sensor. When AFB1 is present, the AFB1-Apt preferentially binds to it and form a stable complex. This complex cannot be adsorbed by SWCNHs, resulting in a fluorescence response. When AFB1 is not present, AFB1-Apt is adsorbed by SWCNHs, resulting in fluorescence quenching. It is worth mentioning that the constructed sensing platform has a simple structure and operation and meets the basic requirements of detection. This makes it low cost and suitable for even non-specialists to rapidly detect AFB1 contamination in food at the field level. Our sensing method has the potential to be further studied and extended to the detection of other contaminants, providing a feasible strategy for rapid detection.

## 2. Materials and Methods

### 2.1. Materials

Aflatoxin B1 (AFB1), aflatoxin B2 (AFB2), aflatoxinG1 (AFG1), aflatoxin G2 (AFG2), ochratoxin A (OTA), zearalenone (ZEA), T-2 toxin, and fumonisin B1 (FB1) were purchased from Qingdao Prebon Biological Engineering Co., Ltd. (Qingdao, China). Soybean oil was purchased from a local supermarket in Changchun, China. AFB1-Apt (5′-FAM-GTTGGGCACGTGTTGTCTCTCTGTGTCTCGTGCCCTTCGCTAGGCCCACA-3’) [35] was synthesized by Shanghai Biotechnology Co., Ltd. (Shanghai, China). Single-walled carbon nanohorns (SWCNHs) were purchased from Xianfeng Nanomaterials Technology Co., Ltd. (Nanjing, China). Tris-HCl buffer (pH 8.8, 1 M) was purchased from Beijing Solarbio Science & Technology Co., Ltd. (Beijing, China), and 1 × TE buffer (pH 7.8–8.2) was purchased from Shanghai Sangong Biotechnology Co., Ltd. (Shanghai, China). A stock solution was prepared by dissolving AFB1-Apt in 0.1 × TE buffer, which was stored in portions at −20 °C. The binding buffer (50 mmol/L Tris-HCl, 10 mmol/L MgCl_2_, 100 mmol/L NaCl, 5 mmol/L KCl, pH 8.0) was used for a binding reaction between the aptamer and AFB1. All other chemicals were of analytical grade and purchased from commercial suppliers and used without further purification.

### 2.2. Instrumentation

The morphology of SWCNHs was analyzed using a SU8220 scanning electron microscope (SEM) (Hitachi, Japan) and a Talos F200X G2 transmission electron microscope (TEM) (Thermo Fisher, Waltham, MA, USA). Fluorescence intensity experiments were performed using a VICTOR Nivo multifunctional enzyme labeler (PerkinElmer, Waltham, MA, USA). The excitation wavelength was set to 495 nm at room temperature, and fluorescence spectra were recorded in the emission wavelength of 500 nm to 620 nm range. The slits used for excitation and emission measurements were set to 5 nm. Fluorescence intensity was recorded at the maximum emission wavelength of 520 nm. Unless otherwise stated, all measurements were performed at room temperature.

### 2.3. Optimization of System Conditions

To obtain the optimal reaction system, relevant experimental parameters, such as SWCNH concentrations, incubation times, and incubation temperatures, were investigated. Among them, the mass ratio of SWCNHs and aptamers (50:1–600:1) was used to optimize the concentration. In addition, two time parameters, i.e., the incubation time of AFB1-Apt with SWCNHs (5–50 min) and the incubation time of AFB1-Apt with AFB1 (30–150 min), were used for optimization. Finally, the optimal temperature for the sensing system was evaluated using 4–45 °C.

### 2.4. Detection of AFB1

AFB1 was analyzed quantitatively under optimal conditions. Firstly, 100 μL of AFB1 aptamer (100 nM) was heated at 90 °C for 10 min, then cooled in ice for 10 min, and then kept at room temperature. After that, 100 μL of different concentrations of AFB1 standards (0–500 ng/mL) were added and incubated at 37 °C for 2 h. Subsequently, an appropriate amount of SWCNH solution was added to the binding buffer in 1.5 mL centrifuge tubes with a final volume of 500 μL, in which the binding buffer was substituted for the target as a blank control group. The solution was then incubated at 37 °C for 30 min and centrifuged at 8791 g for 15 min. The supernatant was used for fluorescence intensity determination. The standard curve was plotted using the target concentration and fluorescence on the horizontal (*x*) and vertical (*y*) axes, respectively. All measurements were repeated three times.

### 2.5. Performance Analysis

To evaluate the various performances of the developed AFB1 sensing assay, several common toxins (AFB2, AFG1, AFG2, OTA, ZEA, T-2, and FB1) were comparatively used to assess their specificity, and six independent aptasensors were prepared to evaluate their reproducibility. In addition, measurements were taken every 12 h to verify their stability (aptasensors were stored at 4 °C before use). The assay procedure involved was the same as the conditions used for the AFB1 assay.

### 2.6. Detection AFB1 in Soybean Oil Samples

To verify the applicability and reliability of the developed fluorescent aptasensor for AFB1 detection, soybean oil was used. Soybean oil was processed according to Li et al. [36] with some modifications. Briefly, AFB1 standard sample was added to 5 g of soybean oil. Then, 25 mL of methanol (60%) was added, and the mixture was homogenized using agitation with vortex at room temperature for 25 min. Then, the homogenized mixture was centrifuged at 4410 g for 5 min, and the supernatant was filtered twice with a 0.22 um filter membrane. Finally, the filtrate was diluted to 10 ng/mL, 50 ng/mL, and 100 ng/mL with binding buffer (pH 8.0). Meanwhile, to verify the accuracy of the developed fluorescent aptamer sensor, an aflatoxin B1 ELISA kit (EKT-010-48/96T) was used to determine AFB1 in soybean oil samples according to the instructions.

## 3. Results and Discussion

### 3.1. Sensing Strategy for AFB1 Detection

In this study, we developed a simple and economical fluorescent aptasensor for the detection of AFB1. SWCNHs can bind to the AFB1-Apt to undergo fluorescence resonance energy transfer resulting in fluorescence quenching, showing different adsorption effects on DNA strands with different structural states (Figure 1). In addition, the use of fluorophore labeling can effectively reduce the impact of foreign probes on their sensing systems [37]. When AFB1 is present, the AFB1-Apt of the modified fluorescent group binds to its specific regions to form a stable secondary structure complex. The complex can resist the adsorption of the SWCNHs, so that the complex and SWCNHs cannot become close to each other. Therefore, they cannot trigger the fluorescence resonance energy transfer to quench the fluorescence signal. Thus, the fluorescence signal in the system is still present in stronger condition. When AFB1 is not present, the AFB1-Apt is in the free single-stranded DNA state and is adsorbed on the surface of SWCNHs. This allows the single-stranded aptamer with FAM to trigger a FRET effect with SWCNHs and quenching of the fluorescence signal. Finally, AFB1 can be detected using the measured fluorescence intensity in the sensing platform.

### 3.2. Feasibility Study

To verify the feasibility of the constructed sensing platform, fluorescence spectroscopy analysis was performed. As shown in Figure 1a, a significant fluorescence response of free AFB1-Apt occurred around 520 nm out, while the fluorescence values were measured for the presence and absence of AFB1. As can be seen from line c, in the absence of AFB1, the aptamer is adsorbed by SWCNHs, leading to the FRET effect of the fluorophore on SWCNHs, and thus, the fluorescence is quenched. In the presence of AFB1, the aptamer and AFB1 bind specifically forming complex, and SWCNHs cannot adsorb the formed complex (Figure 1b). Thus, the fluorophore on this complex is dispersed with SWCNHs, resulting in high fluorescence response values, where the presence of the target AFB1 significantly mitigates the decrease in fluorescence. The results shown are consistent with previous reports [31]. Therefore, the concentration of AFB1 and the change in fluorescence intensity after the reaction are positively correlated, and the change in fluorescence intensity can be used for the quantitative detection of AFB1.

### 3.3. Characterizations of SWCNHs

SWCNHs were morphologically characterized using a SU8220 scanning electron microscope (SEM) and an FEI Talos F200x high-resolution transmission electron microscope (TEM). The SEM image (Figure 2A) shows that the SWCNHs particles are spherical aggregates, and the TEM image (Figure 2B) shows that the SWCNHs are uniformly dispersed with an average diameter of about 50 nm, and the length and diameter of the nanohorns are about 10 nm and 2 nm, respectively [38,39].

### 3.4. Optimization of the System Condition

In the constructed sensing system, different experimental parameters can affect the specific binding of AFB1 to the aptamer, and thus the accuracy of the fluorescence test needs to be optimized. Therefore, relevant experimental parameters, such as SWCNH concentration, incubation time, and incubation temperature were optimized to obtain a better aptasensor performance [40,41].

#### 3.4.1. Optimized SWCNH Concentration

The sensing strategy is mainly based on the FRET effect of fluorescent groups (FAM), SWCNHs, and fluorescence quenching; therefore, the concentration of SWCNHs in this quenching system plays an important role in the whole system [42]. In this work, by studying the fluorescence intensity at different mass ratios of SWCNHs and AFB1-Apt, the errors caused by different aptamer concentrations were minimized, and the maximum quenching efficiency was thus more easily achieved. In this section, the fluorescence quenching rate between different mass ratios of SWCNHs and AFB1-Apt is shown in Figure 3A. The fluorescence value of the supernatant gradually decreases and the fluorescence quenching rate increases with an increase in mass ratio. The fluorescence quenching rate decreases abruptly after the mass ratio of SWCNHs to aptamer increases to 200:1 and stabilizes at 300:1. Therefore, 300:1 (m_SWCNHs_:m_ssDNA_) was chosen as the optimal mass ratio for fluorescence signal bursting of aptamers and was used for further experiments.

#### 3.4.2. Optimized Incubation Time

Incubation time plays a crucial role in the successful construction of the whole sensing platform [25]. To optimize the incubation time of the system, two experimental parameters were optimized in this experiment. Firstly, the incubation time of AFB1-Apt with SWCNHs, and secondly, the incubation time of AFB1-Apt with AFB1.The results of time optimization are shown in Figure 3B, which shows that the fluorescence quenching rate ((F0-F)/F0, where F is the experimental group, F0 is the control group) reached the highest value of 93 % after 30 min incubation of AFB1-Apt and SWCNHs. The fluorescence quenching rate did not fluctuate more in the longer interval. To ensure the maximum fluorescence quenching rate of AFB1-Apt and SWCNHs, 30 min was chosen as the optimized incubation time. Meanwhile, the results of the incubation time for AFB1-Apt with AFB1 are shown in Figure 3C. With the gradual increase in incubation time, the fluorescence value of the supernatant kept increasing, and the relative fluorescence intensity also increased. The relative fluorescence intensity reached to maximum when the incubation time increased to 120 min. To maximize the binding of AFB1-Apt to AFB1 and for the consideration of time cost in the assay method, 120 min was chosen as the optimal reaction time for AFB1-Apt with AFB1.

#### 3.4.3. Optimized Incubation Temperature

The temperature in the detection system is also one of the key factors affecting the binding of the aptamer and AFB1 [43]. To detect the fluorescence intensity in the system at different temperatures, several temperatures were selected in the experiment, including 4 °C, 20 °C, 30 °C, 37 °C, and 45 °C. The results of the optimization of the system temperature in this work are shown in Figure 3D. The relative fluorescence intensity was lowest at 4 °C and 45 °C, while the maximum was at 37 °C. The lowest fluorescence under a low-temperature environment of 4 °C may be due to the slow diffusion of reactants in the system, which could not fully react within the specified time. The high temperature of 45 °C may have a negative effect on the whole reaction system and may affect the binding of the aptamer and AFB1. In contrast, within the suitable environment of 37 °C, the SWCNHs are better dispersed, and it is easy to obtain a better quenching effect compared with other temperatures, which is more favorable to the binding of the aptamer and AFB1. Therefore, 37 °C was chosen as the optimal temperature for the whole reaction system in the assay system established in this experiment.

### 3.5. Detection Performance of the AFB1 Aptasensor

To evaluate the performance of the proposed aptasensor, the fluorescence intensity of the detection system was measured under optimized conditions for the detection of different concentrations of standard AFB1. The standard AFB1 (0–500 ng/mL) was incubated with the aptamer, and the detection system was used for fluorescence measurements. As shown in Figure 4A, the relative fluorescence intensity (minus the peak of the blank control) gradually increased with increasing AFB1 concentration. In addition, Figure 4B illustrates the linear relationship between fluorescence intensity and AFB1 concentration in the range of 10–100 ng/mL, where the regression equation for AFB1 is Y = 7.1104X_[AFB1]_ + 98.39754 (R^2^ = 0.99202). In this regression equation, Y represents the relative fluorescence response in the assay system, X represents the concentration of AFB1, and R^2^ refers to the goodness of fit, which represents how well this regression equation fits the observed values. The limit of detection (LOD) was calculated to be 4.1 ng/mL based on the triple standard deviation of the blank response (3SD/slope, where SD is the standard deviation of the control sample and slope is the standard curve’s slope). In addition, the designed fluorescent aptamer sensor was compared with other methods reported in the literature for the detection of AFB1. As shown in Table 1, the fluorescent sensing platform has excellent performance for AFB1 detection compared to the other detection methods. Also, the sensing platform demonstrated comparable detection limits compared to other superior signal amplification strategies and was below the maximum allowable levels of 8–20 μg kg^−1^ recorded in the EU and several countries [44,45]. Notably, the designed sensing platform is easier to construct, with simpler steps and lower cost. The above results confirm that this simple fluorescent aptasensor based on SWCNHs is feasible and sensitive for the detection of AFB1.

### 3.6. Performance Analysis

In addition to sensitivity, selectivity is an important parameter of the sensing platform. In the actual detection environment, there are often several toxins coexisting, and cross-reactivity between different toxins and aptamers may occur. To determine the selectivity of the method, the influence of other fungal toxins, including AFB2, AFG1, AFG2, OTA, ZEA, T-2, and FB1, on the analytical method were investigated [45]. In the specificity experiments, the concentration of AFB1 was 100 ng/mL, while the concentration of other toxins was 500 ng/mL. The the specificity of the aptamer was effectively verified, and the selectivity of the sensing strategy for AFB1 was more instinctively seen. As shown in Figure 5, the addition of AFB1 resulted in a significant increase in the intensity of the fluorescence associated with the detection system. The addition of other types of toxins as well as congeners resulted in fluorescence values that were much lower than the fluorescence values of AFB1. The experimental results illustrate that in the presence of AFB1, a stable secondary structure complex is formed by AFB1 with AFB1-Apt that is not adsorbed by SWCNHs, resulting in enhanced fluorescence values. Therefore, this experiment established an aptamer sensing detection system with good specificity for AFB1.

To assess the reproducibility of the aptasensor, six independent aptasensors were prepared using the same method to detect the same concentration of AFB1 (100 ng/mL) (Figure 6A). Similar fluorescence responses were obtained, and the relative standard deviation (RSD) was calculated, which was 5.3%, all of which demonstrated the good reproducibility of the aptamer sensor. In addition, its stability (0–72 h) was studied. The prepared sensors were stored at 4 °C prior to the assay and tested once in 12 h. The results are shown in Figure 6B, where the aptamer sensor retained 82.9% of the initial fluorescence signal response after 48 h of storage, confirming the excellent storage stability of the prepared aptamer sensor.

### 3.7. Analysis of AFB1 in Soybean Oil

The annual proportion of mycotoxin contamination in global food crops, specifically, AFB1 in soybeans, is about 25% [53]. To evaluate the practicality and reliability of the aptamer sensor for the analysis of real samples, soybean oil was spiked, in which AFB1 was added at concentrations of 10 ng/mL, 50 ng/mL, and 100 ng/mL. The analytical results, as listed in Table 2, showed that the recoveries of this aptamer sensor were in the range of 85.9%~102.3% with RSDs of 3.59%~5.46%. The results obtained were compared with the results obtained using commercial ELISA assays and showed little difference in the recoveries and, in some cases, a smaller RSD. In addition, this strategy also provides an effective method for the detection of AFB1 and holds good promise for expansion to the detection of other toxins.

## 4. Conclusions

In this study, we designed a simple, economical, and reliable fluorescent aptasensor for the successful detection of AFB1 in food. SWCNHs were introduced as novel quenchers, and FAM-modified AFB1 aptamers were used as fluorescent probes in this sensing strategy. An analysis under the optimized parameters showed that the aptasensor responded well to AFB1. Under the optimized conditions, the fluorescence intensity increased linearly with the AFB1 concentration in the range of 10–100 ng/mL, and the detection limit was as low as 4.1 ng/mL. In addition, the sensing method successfully detected AFB1 in soybean oil samples, demonstrating the practical feasibility of the method and the great potential of aptasensors in the field of food safety. Based on the above advantages, the proposed sensing method provides an economical and rapid detection strategy for the quantitative detection of AFB1. In future studies, our sensing method can hopefully be extended to detect other contaminants, providing a promising strategy for rapid field detection.

## Data Availability

Raw data can be provided by the corresponding author upon request.

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
