# Peer review of "Single-Walled Carbon Nanohorn-Based Fluorescence Energy Resonance Transfer Aptasensor Platform for the Detection of Aflatoxin B1"

_foods, 2023, doi:10.3390/foods12152880_

Round 1

Reviewer 1 Report

The manuscript by Fan et al. describes an aptamer-based method for aflatoxin B1 detection. The method is based on AFB1-specific fluorescently labelled aptamer and single-walled carbon nanohorns (SWCNHs) that function as fluorescence quenchers upon aptamer adsorption. AFB1 bound aptamer forms a stable complex which prevents the SWCNH adsorption and subsequent fluorescent quenching. The described method is an interesting and simple alternative for chromatographic and immunoassay technologies for aflatoxin detection which is undoubtedly an important research topic. However, the manuscript lacks novelty, and the methods are not described with enough detail. For example, it is not described where or how the SWCNHs, one of the main components of the assay, were obtained. The paper by Qi el al from 2022 (doi.org/10.1007/s00216-022-03938-3) that is also cited in this manuscript describes a very similar aptasensor for aflatoxin detection as well. The authors’ claim that in this manuscript “a novel and rapid fluorescent aptamer sensing assay were constructed for the rapid detection of AFB1 in combination with SWCNHs for the first time” (lines 85–86) should be therefore revised. The authors should clarify what has been done before in field of aflatoxin detection as well as aptasensors to better elaborate what in fact is the novelty of their method in comparison with existing assays and sensors. It is not clear for the reader why this aptamer-based assay would solve the issues of other existing methods and the described disadvantages of immunoassays lack credibility (i.e. the authors’ claims are not supported by experimental data or literature references). Therefore, the manuscript requires major revision. Some more detailed major and minor comments are described below.

Major comments:

Spiking of samples: Why did the authors spike the oil after the sample treatment? It would be more like a real sample if the sample was spiked before the sample handling. How do you otherwise know that the extraction method works?

Experimental part: Please include all the details of the assay procedure with the required detail so that another scientist is able to reproduce the assay. Please include in the materials section how the SWCNHs were manufactured (or where they were purchased).

Why is the centrifugation step needed before the measurement? Is it not so that the SWCNHs are then separated in a pellet and the supernatant then does not include the SWCNHs nor the aptamer bound to them? Therefore, if the fluorescence of the supernatant decreases it is not a question of fluorescence quenching but due to the fact that SWCNH-bound aptamers are not present in the supernatant.

Line 52: Please elaborate the claim of inadequacy of antibodies and support the claim with appropriate references.

Lines 130–156: Important details are missing from the assay description. What do you mean with pre-treated aptamer solution? What was the reaction volume? Which buffer was used? Please wire the centrifuge speed in g force not rpm as this depends on the rotor. How did you do the incubation, in a well plate, in cups, in a cuvette, please specify. What kind of microfiltration membrane was used? What was the molarity and pH of the Tris-HCl buffer. Which ELISA kit was used? Please describe the entire protocol at the level that someone could repeat it.

Lines 222–223: Why does the fact that the mass ratio increases to 200:1 indicate that the fluorescence of AFB1-Apt is quenched by SWCNHs? Please revise the chapter to clarify the mechanism of the fluorescence quenching.

Lines 231–234: Is fluorescence quenching rate different from fluorescence recovery rate? What about quenching efficiency? Please describe what these terms mean to clarify it to a reader who might not be that familiar with the topic.

Line 272: Is the x-axis of Figure 4B log AFB1 concentration as described on this line, or linear as in the actual figure? This is a bit confusing.

Line 275: What do you mean with X represents the concentration of the dual mycotoxin?

Lines 282–285 and Table 1: The authors claim that the designed sensing platform is easier to construct with simpler steps and lower cost. Please include more details to the text and table (e.g. column such as “analysis time”, “analysis steps” etc). Also, please add some discussion about the current sensitivity requirements for AFB1 analysis. Is the LOD of this method suitable for aflatoxin analysis in food samples? Does it meet the regulatory limits?

Lines 297–311: What about cross-reactivity with other aflatoxins? Does the aptamer and the sensing system detect only AFB1 or also other aflatoxins, such as commonly existing ones AFB2, AFG1, AFG2, AFM1? At least the European regulation (COMMISSION REGULATION (EC) No 1881/2006) has set maximum residue limits for AFB1 alone as well as for the sum of AFB1, AFB2, AFG1, AFG2, and therefore, would be important to know whether the sensing system is able to detect AFB1 alone or all of them.

References: Please double check all the references for correct capitalization (e.g. line 402 should be B1 and T7, not b1 and t7, and line 408 should be PVP and SYBR, not pvp and sybr)

Mostly the English language is sufficiently good but please revise the text for further clarity and some grammar mistakes.

Reviewer 2 Report

This paper reported a single-walled carbon nanohorns-based fluorescence energy resonance transfer aptasensor platform for detecting aflatoxin B1. The study involved the Fluorescent moiety (FAM)-modified aptamers sensing platform. I suggested comments below;

1. This manuscript has some spelling and typos errors. Abbreviations should be checked to describe when they are used the first time.

2. Relevant and important literary work in the field can be added in the manuscript to enhance the quality and readership of the manuscript. In the introduction section, some recent references related to the CNT based sensors should be provided. For example: J. Electrochem. Soc. 167, 167519, 2020.

3. The manuscript should include an explanation of the aflatoxin B1 detection mechanism and highlight the novelty of the work.

4. How LOD was calculated.

5. Figure 5, the sensor showed responses for the other common fungal toxin when measured separately. However, when mixed together they showed almost the same response compared to only AFB1.

6. Stability and reproducibility are the two crucial parameters of any sensor. These should be checked and included in the revised manuscript.
